# A unified pipeline for FISH spatial transcriptomics

## Graphical abstract

## Authors

Cecilia Cisar, Nicholas Keener,
Mathew Ruffalo, Benedict Paten

## Correspondence

ccisar@ucsc.edu

## In brief

Spatial transcriptomics is a rapidly growing field, but there is a lack of standardized tools for analyzing high-resolution experiments, leading many groups to write their own in-house tools. To address this, Cisar et al. have developed PIPEFISH, a semi-automated and generalizable pipeline for performing transcript annotation for FISH-based spatial transcriptomics.

## Highlights

- PIPEFISH is a pipeline tool for FISH spatial transcriptomics data processing

- Spatially annotated transcripts can be extracted from FISH images with minimal setup

- Internal quality metrics can be used to judge pipeline performance

- PIPEFISH finds transcripts matching orthogonal measures of expression in real data

Cisar et al., 2023, Cell Genomics 3, 100384
September 13, 2023 © 2023 The Authors.

CellPress

## Article

# A unified pipeline for FISH spatial transcriptomics

Cecilia Cisar,[1,3,4,*] Nicholas Keener,[1,3] Mathew Ruffalo,[2] and Benedict Paten[1]
[1]Department of Biomolecular Engineering, School of Engineering, University of California, Santa Cruz, Santa Cruz, CA 95064, USA
[2]Computational Biology Department, School of Computer Science, Carnegie Mellon University, Pittsburgh, PA 15213, USA
[3]These authors contributed equally
[4]Lead contact
*Correspondence: ccisar@ucsc.edu

## SUMMARY

High-throughput spatial transcriptomics has emerged as a powerful tool for investigating the spatial distribution of mRNA expression and its effects on cellular function. There is a lack of standardized tools for analyzing spatial transcriptomics data, leading many groups to write their own in-house tools that are often poorly documented and not generalizable. To address this, we have expanded and improved the starfish library and used those tools to create PIPEFISH, a semi-automated and generalizable pipeline that performs transcript annotation for fluorescence *in situ* hybridization (FISH)-based spatial transcriptomics. We used this pipeline to annotate transcript locations from three real datasets from three different common types of FISH image-based experiments, MERFISH, seqFISH, and targeted *in situ* sequencing (ISS), and verified that the results were high quality using the internal quality metrics of the pipeline and also a comparison with an orthogonal method of measuring RNA expression. PIPEFISH is a publicly available and open-source tool.

## INTRODUCTION

Development of single-cell RNA sequencing (scRNA-seq) over the past decade has allowed researchers to probe the heterogeneous nature of real tissue by characterizing the transcriptome of individual cells. This has led to the discovery of many new cell types and has enhanced our understanding of the mechanisms of disease.[1] However, in the process of separating single cells from each other so that their transcriptomes can be sequenced individually, the spatial context of each cell and the location of each transcript within cells, both of which contain important biological information, are lost. Spatially resolved transcriptomics methods can be used to characterize transcriptomes on a single-cell level such that the spatial context of each transcript and cell is also recovered. This spatial information is incredibly useful biomedically, as many diseases can be characterized by abnormal spatial patterns,[2] and in developmental biology, where many vital and early processes are driven by spatial relationships between different biological factors[3] and defects in their spatial distributions can have serious consequences for the developing organism.

Named as *Nature*'s "Method of the Year" in 2020,[4] spatially resolved transcriptomics is becoming increasingly prevalent in new published research after the first multiplexed methods were demonstrated in the mid 2010s,[5,6] with new methods and discoveries being released at an ever increasing pace. The earliest spatial transcriptomics methods were based on fluorescence *in situ* hybridization (FISH) probes with specific mRNA targets, which

could then be imaged with a microscope.[7] These early methods were very low throughput, only targeting a handful of genes,[8] but more recent advances in multiplexing these methods have allowed for nearly whole-transcriptome-level targeting. Exposing samples to many sequential rounds of hybridization with different sets of FISH probes, called seqFISH,[9] was used to go from a few dozen target genes to hundreds and has been shown to be effective in both cell cultures[10] and in tissue samples.[11] A newer version, seqFISH+,[12] has shown that it is possible to characterize the expression of over 10,000 different mRNA targets simultaneously. A similar approach is used in MERFISH but with more advanced built-in error correction abilities, allowing for increased accuracy of results.[13] Another *in situ* imaging-based technique, called *in situ* sequencing (ISS), takes advantage of short-read sequencing technology to image the location of mRNA targets as they are sequenced using traditional sequencing-by-synthesis technology.[5] Experiments based on this method have been able to generate spatial maps of breast cancer tumors that could be useful in clinical diagnostics of patients with cancer, leading to improved patient outcomes.[14] Advances in ISS methods have led to increased throughput and accuracy.[15]

As these spatial transcriptomics methods further improve to become higher resolution and throughput while also becoming less complex to perform, the number of researchers who wish to use them will also increase. Thus, it is also important to have available standardized, open-source methods for processing and analysis of the results from these spatial methods for the purpose of reproducibility and easy comparison of results

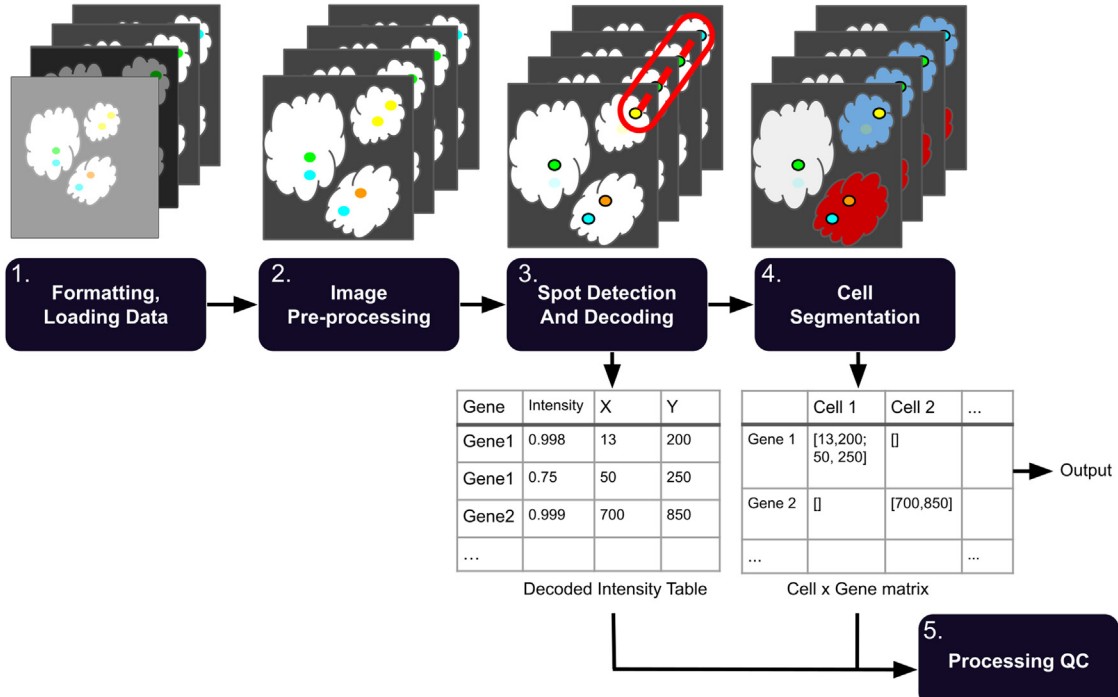

**Figure 1. Overall schematic of processing pipeline**

The workflow divides the processing into multiple discrete steps. (1) Raw images are optionally sorted into pseudorounds and pseudochannels and then converted to the standardized SpaceTx format (STAR Methods). (2) Images undergo pre-processing such as registration, white tophat filtering, high-pass and low-pass Gaussian filters, and histogram matching. (3) Transcripts are identified within each image and listed in a table with (x,y,z) coordinate information. (4) Cell boundaries are identified and transcripts are assigned a cell ID number. (5) QC metrics for the experiment results and processing are calculated from the decoded output.

between different research groups. Much of the analysis done for published research relies on in-house code with little documentation, making the adoption of these experiments difficult and time consuming. Existence of publicly available, open-source computational tools for analysis of spatial data will make these types of experiments more accessible for labs that wish to perform them, facilitating the generation of a larger body of spatial data.

Although some current sequencing-based spatial transcriptomics methods have open-source libraries[16,17] or bundled toolkits[18] available, FISH-based methods have significantly fewer options at their disposal.[19] Experimental apparatus must usually be assembled in-house, and most labs write their own analysis code. Written by the Chan-Zuckerberg Institute, starfish[20] is an open-source Python library containing methods and data types for the processing and analysis of FISH-based spatial imaging results. It is capable of taking raw image files and outputting decoded results detailing the spatial location of each transcript, which cell it is located in, and where that cell is located in the original image. These results can then be used in downstream tools such as squidPy[21] or stLearn.[22] While useful, starfish is lacking in several different areas, including requiring a large number of input parameters to carry out a full analysis, having only several basic image pre-processing tools, lacking an adequate seqFISH decoding algorithm, and the absence of quality control metrics to determine the significance of the results.

We have created a universal spatial transcriptomics pipeline for FISH-based methods called PIPEFISH using the CWL (Common Workflow Language) framework and an improved version of the starfish package that includes a novel seqFISH decoding method and PoSTcode[23] as an ISS decoding option. PIPEFISH organizes the tools available in our custom starfish module into an ordered pipeline that can take raw image data from the most popular FISH-based methods and extract spatially annotated mRNA transcript counts from them. We have also developed a number of quality control metrics that can be used to assess the performance of the pipeline results. Using the pipeline, we found spatially annotated transcripts from three different *in situ* image datasets, seqFISH of a mouse embryo, MERFISH of human U2-OS cells, and targeted ISS in a mouse brain, and used both the internal QC (quality control) metrics and an external metric to show that our results are high quality.

## RESULTS

### A general SpaceTx pipeline

We have developed PIPEFISH, a general pipeline to extract transcript locations from the raw results of a FISH experiment (Figure 1). Input can be accepted in many forms so long as images are validly encoded tiffs and a codebook describing the expected transcripts is provided. The codebook is expected to have transcript names and their paired "barcodes,", which is

the list of imaging round and color channel combinations where the transcript is expected to fluoresce. Additional information can be provided in a configuration json file with experimental design details, such as the hamming distance between barcodes if the experiment was designed such that incomplete barcodes can be recovered during decoding.

The configuration json file can also include parameters for running image processing, such as detailing which auxiliary views (if any) should be used for image registration or subtracted from the primary views as a background. Other tunable parameters for image processing, such as the kernel radius for rolling ball background subtraction or a white tophat filter, can also be specified.

There are two primary types of image decoding: pixel based and spot based. The pixel-based method treats each intensity at a given (x,y,z) location as a value in a vector across all rounds and channels and then assigns the barcode (if any) most likely to match that vector. The spot-based method first identifies spots in the image as local intensity peaks and then applies one of several "decoding" methods to predict which spot locations correspond to transcripts.

Transcripts can then be assigned cell IDs by applying a segmentation mask. There are naive methods included to generate a mask from a provided auxiliary view, using thresholding and watershed based approaches or using the neural-network-based CellPose segmentation tool.[24] If using CellPose, the user can choose one of the pre-trained models offered by CellPose or they can use their own custom model. External masks, such as those produced by Fiji or Ilastik, can also be imported.

Should troubleshooting of an individual step be needed, the pipeline has been configured to allow for each step to be run individually from prior output and the same json configuration file. We hope to address the needs of all FISH experiments in this comprehensive and simple-to-use pipeline.

### An improved seqFISH decoder
We found that starfish decoding of seqFISH spots using starfish's decoding methods consistently produced far fewer mRNA targets than expected, which can reduce the accuracy of many downstream analyses. This is a result of the starfish's seqFISH decoders requiring that spots must be at least mutual nearest neighbors of each other to be connected into a barcode, which is very sensitive to spot drift, a common problem in seqFISH experiments with high numbers of hybridizations. To address this, we developed the CheckAll decoder, which considers all possible spot combinations that could form barcodes and chooses the best non-overlapping set (Figure S1).

The CheckAll decoder checks all possible barcodes that could be formed from the given spot set and chooses those most likely to represent true mRNA targets (STAR Methods). It is based on the method used by the original seqFISH authors[9] but features some improvements and an additional option that allows for adjustment of the precision/recall tradeoff. This "mode" parameter can take three different values and controls this tradeoff by setting several parameters that are not under user control, with high accuracy mode resulting in higher precision but lower recall, low accuracy mode having lower precision but higher recall, and the medium accuracy mode falling somewhere in between. Also, unlike the current starfish decoders, the CheckAll decoder is capable of decoding error-corrected barcodes. If each barcode used has a hamming distance of at least two from every other code, they can be uniquely identified even without a complete barcode. These error-corrected transcripts found with an incomplete barcode are less accurate (more prone to false positives) than those decoded with complete barcodes but can significantly increase recall if that is preferred.

### QC metrics
In order to provide qualitative confidence in the results from this pipeline, we defined a set of internal QC metrics that can be applied to the vast majority of experiment types. These QC results are automatically generated during a pipeline run and provide a variety of statistics and graphs for each provided field of view (Figures S2 and S3). Some of these metrics rely on "off-target" barcodes that do not correspond to actual experimental probes included; these can be added to the codebook manually or can be inserted automatically during the pipeline's initial conversion step. Off-target barcodes are treated identically to normal barcodes during decoding and are regarded as false positives in the QC step, something that has been experimentally verified to be functionally identical to including false positive probes that do not match any transcripts.[13] We have found that these off-target-based metrics are particularly useful for directly comparing sets of results (Figure S2), as the false positive rates and the threshold of true decoded barcodes that can be delineated from false positive barcodes give a quantifiable point of comparison between datasets. Transcripts identified through the error correction method, where the barcode does not exactly match a known target and is still close enough to be uniquely identified, tend to be more prone to false positives than non-corrected transcripts. Because of this, we show results both with and without error-corrected barcodes in the relevant QC figures to allow users to determine whether they would like to use the error-corrected transcripts in their analyses.

By comparing these QC results across different pipeline parameters, it is possible to determine which set of results is the highest quality. Some of the additional metrics (Figure S3; STAR Methods) may help to troubleshoot which pipeline stage needs altered parameters. The inclusion of these metrics to the pipeline allows for iterative parameter selection to get the best results possible from a set of input data.

### Pipeline performance on real FISH datasets
To show that the tools we have built are capable of obtaining high quality results from a variety of FISH experiment types, we ran PIPEFISH on publicly accessible data from three different FISH experiments, each of a different type: seqFISH of a mouse embryo[11] (351 genes), targeted ISS of a mouse brain[23] (50 genes), and MERFISH of human U2-OS cells[25] (130 genes). After transcript locations were obtained, we evaluated QC metrics, using both the aforementioned methods internal to PIPEFISH and the orthogonal methods external to PIPEFISH, to evaluate performance.

#### Internal quality check
For each dataset, we initially evaluated performance using the internal QC metrics produced by the pipeline. One set of metrics that can be useful for all datasets are those that estimate the background signal through the quantification of off-target

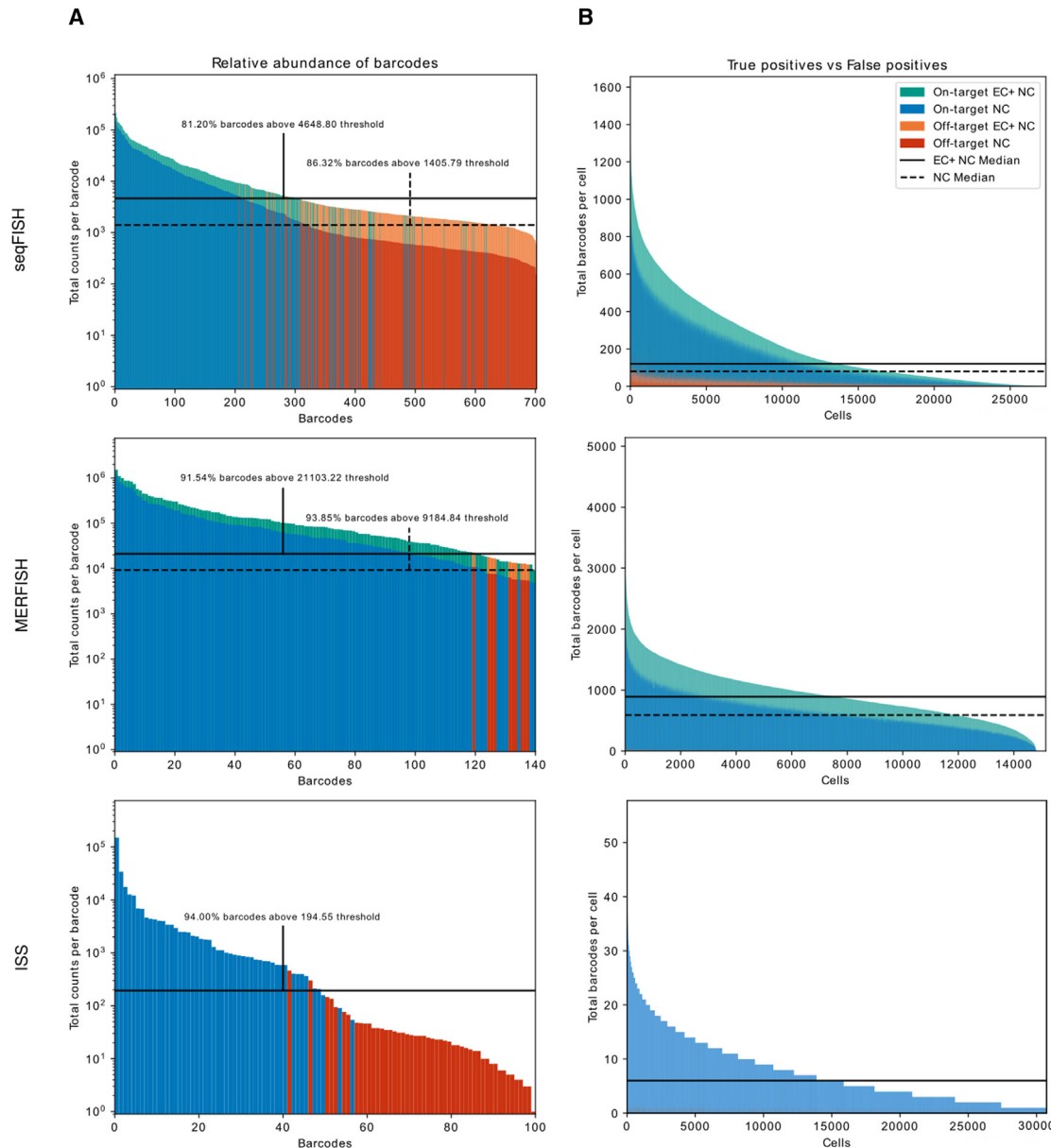

**Figure 2. Quality control figures highlighting false positive off-target barcodes from pipeline output across all fields of view of each dataset**
(A) Total counts of each barcode. Total count of each barcode colored by barcode type, and error-corrected count shown above non-corrected counts (in orange or green) where applicable. Proposed minimum barcode count threshold is calculated as the upper end of the 95% confidence interval of a normal distribution with the mean and standard deviation of the observed off-target tallies. This is calculated for non-corrected barcodes (dashed) and the combination of non-corrected and corrected barcodes (solid).
(B) Transcript count per cell, with the same color scheme as (A). Median transcript count per cell for ISS, seqFISH, and MERFISH. EC, error corrected; NC, non-corrected.

barcodes inserted into the provided codebook. While this method is beholden to how densely packed the codebook is, it provides insight into the estimated false positive rate of the experiment, which single handedly can be criteria to rerun the experiment. Our results show that true on-target barcodes are found at a significantly higher frequency than the off-target codes (Figure 2), indicating that there are likely to be few false positive transcripts in these results.

Additionally, we verified that the novel CheckAll decoder we developed outperforms the native decoders from starfish. We only compare results from starfish's NearestNeighbor decoding method, as the ExactMatch method could only find a handful of targets in our images. Performance was evaluated using the false barcode metric to show that the CheckAll decoder was less prone to false positives while also finding more mRNA targets (Figure S4; Table 1). The cost for this increase in performance compared to the

**Table 1. Summary of CheckAll decoder performance**

| | starfish NN | CA - high | CA - high (% difference) | CA - medium | CA - medium (% difference) | CA - low | CA - low (% difference) |
|---|---|---|---|---|---|---|---|
| Precision (NC) | 0.925 | 0.965 | +4.23 | 0.927 | +0.22 | 0.896 | −3.19 |
| Total on target (NC) | $2.36 \times 10^6$ | $4.14 \times 10^6$ | +54.77 | $4.91 \times 10^6$ | +70.15 | $5.32 \times 10^6$ | +77.08 |
| Precision (NC + EC) | 0.925 | 0.919 | −0.65 | 0.847 | −8.80 | 0.826 | −11.31 |
| Total on target (NC + EC) | $2.36 \times 10^6$ | $6.03 \times 10^6$ | +87.49 | $8.32 \times 10^6$ | +111.61 | $8.65 \times 10^6$ | +114.26 |

starfish NN, CA - high, CA - medium, and CA - low columns show precision and total on-target counts for the starfish NearestNeighbor decoder, CheckAll decoder (high-accuracy mode), CheckAll decoder (medium-accuracy mode), and CheckAll decoder (low-accuracy mode), respectively. Percentage difference columns show the performance difference between the starfish NearestNeighbor decoder and the CheckAll decoder for each accuracy mode. The starfish NearestNeighbor decoder is not capable of detecting error-corrected targets, so NC and NC + EC values are identical. NN, NearestNeighbor; CA, CheckAll; NC, not corrected; EC, error corrected.

starfish decoders comes in the form of increased run times and memory requirements. While the starfish decoders are nearly instantaneous and require very little memory, the CheckAll decoder can take much longer and may require more memory to return results depending on the input (Figure S5). The CheckAll decoder is capable of taking advantage of Python multiprocessing features in order to mitigate the long run times at the cost of additional memory.

### External quality check

In addition to the internal metrics shown, we validated the results given by the pipeline using a form of orthogonal validation for each dataset. For the mouse embryo seqFISH dataset, in addition to the 351 genes measured by seqFISH, there were an additional 36 genes imaged by smFISH that were used to verify predicted expression of genes not measured by seqFISH in the original study.[11] We used the Python package Tangram[26] to predict expression per cell for the 36 genes measured by smFISH using seqFISH counts from the pipeline and cell-type-annotated scRNA-seq counts from the Mouse Gastrulation Atlas[27] and compared this with the actual expression measured by smFISH in the same cells (Figure 3A; STAR Methods). To achieve high-accuracy matching and prediction of unmeasured genes, the seqFISH counts must be accurate, and so we can use the accuracy of the predicted expression, using the smFISH counts as a truth set, as a quality metric to assess the performance of the pipeline at identifying transcripts in the images. The average Pearson correlation between the predicted and measured counts for the 36 smFISH genes was found to be 0.49 (Figure S6). The large range of correlation coefficients (0.12–0.85) is likely a result of differing image qualities between the smFISH genes. As the smFISH images are not multiplexed, they are very sensitive to the choice of threshold when identifying spots in the images. Thresholds for each gene were determined by plotting the number of spots found at a large range of thresholds and calculating the elbow point of the curve. The area under the curve (AUC) can also be used as a rough estimate of how much the intensities of the true fluorescent signal and the noise in an image overlap, and large overlaps would make separating signal from noise more difficult and less accurate. We found that the correlation between predicted and smFISH counts to be fairly strongly correlated with the AUC of the spot count vs. threshold curve with a Pearson correlation of −0.67 (Figure S7), indicating that the genes where performance is low are likely a result of poor separation between signal and noise intensities in that image and not poor-quality seqFISH counts.

As the MERFISH images used here were of a cell culture of a single cell type, we could directly compare the average expression per cell from the pipeline with RNA expression values obtained using some other standard method. We obtained FPKM measurements calculated from bulk RNA-seq experiments of the same cell type as used in the MERFISH images[25] (personal communication, J. Moffitt) and compared these values with the average expression of the cells in the MERFISH images and found them to be highly correlated with a Pearson correlation coefficient of 0.85 (STAR Methods) (Figure 3B), indicating that the counts the pipeline produced are similar to those obtained by bulk RNA-seq.

There exist abundant expression maps of the mouse brain, which allowed us to compare the pipeline results on the targeted ISS images with a trusted reference. We downloaded single gene expression maps from similar sections of mouse brains from the Allen Brain Atlas and found that the spatial distribution of transcripts from the pipeline and the reference atlas images matched for most genes (STAR Methods; Figures 3C and S8). Out of the fifty genes probed, thirty-four had coronal section images in the Allen Brain Atlas Mouse Brain reference, and nearly all of them showed a strong match between the spatial expression patterns found by our pipeline and in the reference data.

### DISCUSSION

As spatial genomics continues to become more popular as an investigation tool, there will be a growing need for computational tools capable of extracting useful information from the raw image data generated by these methods. We have developed this general and customizable pipeline for those who wish to analyze FISH-based spatial transcriptomics images as part of their research. The pipeline's versatility allows it to be used for any experiment involving targeted barcoding of transcripts and includes options for pre-processing image data, a novel seqFISH decoder, and automated quality metrics that can be used to assess performance. Using real data from three different *in situ* methods and a variety of quality metrics, we showed that the pipeline can be used to obtain high accuracy results that can then be used in a number of downstream applications.

The pipeline is currently capable of obtaining annotated transcript locations from raw images with high fidelity but all subsequent analysis is focused on assessing the quality of the results and not on answering any specific biological question. A future

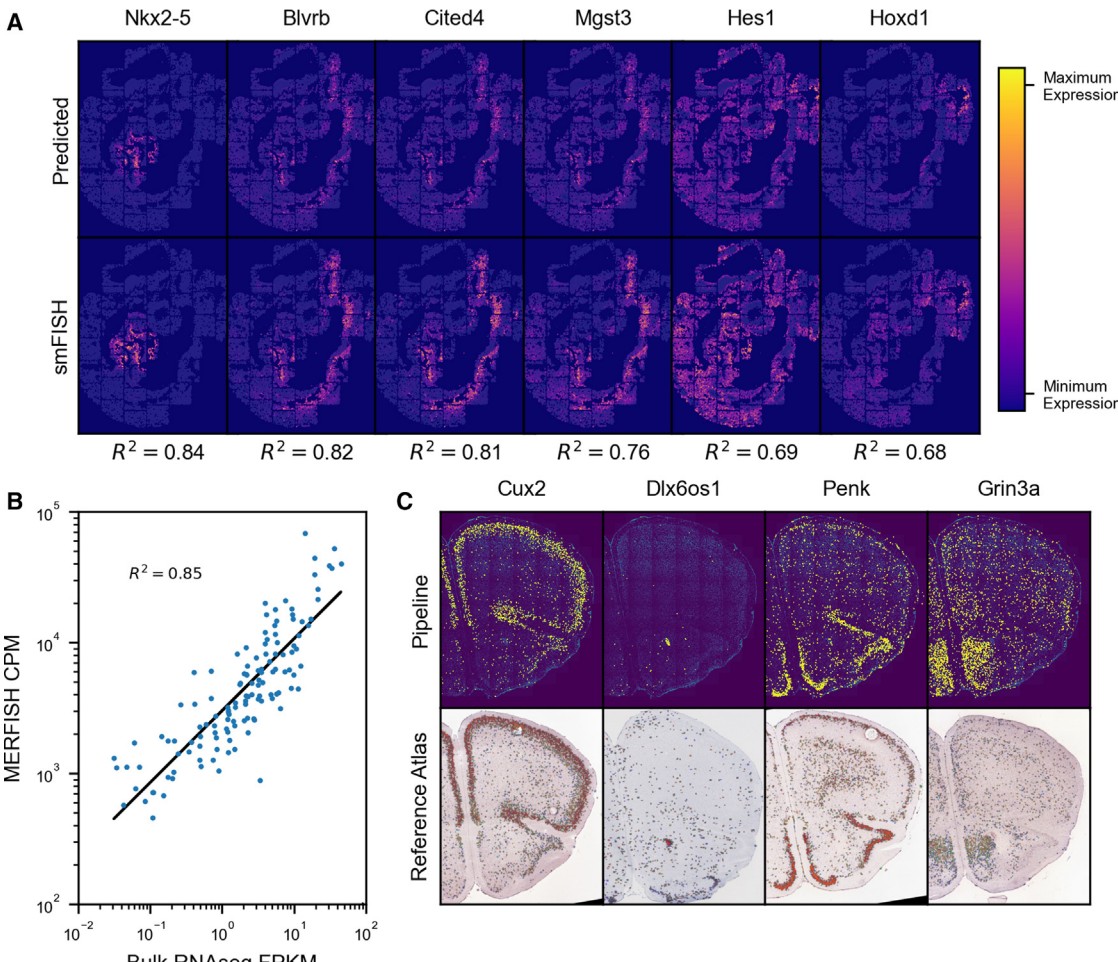

**Figure 3. External validation of pipeline results**

(A) Comparison of predicted gene expression by Tangram with measured expression of the same gene by smFISH (top six correlating genes shown). Each column corresponds to a gene, with the Tangram predicted expression above and the measured smFISH expression below. Pearson correlation coefficient is printed below each.

(B) Correlation of MERFISH counts with bulk RNA-seq FPKM values.

(C) Comparison of spatial distribution of four genes found in the targeted ISS dataset by the pipeline with reference expression of the same genes in a similar brain slice from the Allen Brain Atlas. In the top row, each yellow dot represents a transcript, while the image underneath is the DAPI stain of the sample; in the bottom row, blue dots represent low expression, while more red dots represent higher expression (no color map provided by Allen Brain Atlas), while the image underneath is the Nissl stain of the sample.

version of the pipeline may include options for downstream analyses that could be automatically performed after the transcripts have been decoded and assigned to cells. This could include imputation of missing gene expression counts based on orthogonal scRNA-seq data, cell-type assignment, identification of spatially variable genes, cell neighborhood/communication effects, and general spatial statistics on both the gene and cell level. There already exists a number of tools for any of these analyses, so this would involve testing each to find those worth including. This would make the pipeline an end-to-end workflow for FISH-based spatial transcriptomics analysis, streamlining what is typically a difficult and confusing process.

Here, we have developed a multistage pipeline composed of a series of carefully chosen image-processing algorithms. In the future, it may be possible to use curated results of this pipeline to train a neural-network- or similar machine-learning-based approach to predict the presence of transcripts from FISH images in a single pass or with much less pre-processing. Such a generalized learning approach could integrate additional sources of information to make more accurate predictions, such as cellular location and neighborhood, or to simultaneously predict cell segmentation and could potentially be made computationally efficient through the use of hardware acceleration. We therefore view this work as an important prerequisite to future, more generalized inference on this rich source of data.

## Limitations of the study

PIPEFISH can be used to significantly simplify the process of identifying transcripts in multiplexed FISH images. However, there are non-insignificant hard-drive and RAM requirements in

order to successfully run the pipeline. As each step is run on all FOVs (fields of view) before proceeding to the next step, the output for each step must be saved. Each individual image tile will be duplicated up to three times in the pseudosort (optional), conversion to SpaceTx format, and pre-processing steps, resulting in a large cost to hard-drive space, as multiplexed FISH images can already be quite large. Each FOV could instead be run through the entire pipeline before moving on to the next FOV, only saving the final results, in order to reduce this storage requirement, though this would make it impossible to restart the pipeline at a specific step during troubleshooting. A future version of PIPEFISH could include an option to choose between saving the output from every step and discarding all non-final results so that the user can choose which option suits their needs best. In addition to requiring large amounts of hard-drive space, certain PIPEFISH operations can also be quite RAM intensive, such as the CheckAll decoder, which can require 30 GB+ RAM to decode 2.5 million spots (Figure S5D). Also lacking from PIPEFISH is support for parallelization between FOVs. While some individual operations in PIPEFISH make use of CPU parallelization, it is currently not possible to process more than a single FOV simultaneously. This can result in long run times when there are a large number of FOVs. Toil[28] is a pipeline management system that can execute CWL jobs, and a planned update to PIPEFISH will parallelize the decoding step when run with Toil.

## STAR★METHODS

Detailed methods are provided in the online version of this paper and include the following:

- KEY RESOURCES TABLE
- RESOURCE AVAILABILITY
  - Lead contact
  - Materials availability
  - Data and code availability
- METHOD DETAILS
  - Conversion to SpaceTx format
  - CheckAll decoder
  - Quality control metrics
  - Addition of noise for QC comparison
  - Pipeline parameters used for results
  - smFISH processing
  - Spot count curve as a noise metric
  - Comparison of MERFISH results to bulk RNA-seq
  - Comparison of ISS results to reference atlas

## SUPPLEMENTAL INFORMATION

## ACKNOWLEDGMENTS

This work was supported in part by grants from the National Institutes of Health, grant numbers R01HG010485, U24HG010262, U24HG011853, OT3HL142481, U01HG010961, and OT2OD033761.

## AUTHOR CONTRIBUTIONS

Software, C.C. and N.K.; validation, C.C. and N.K.; formal analysis, N.K.; investigation, C.C. and N.K.; resources, B.P. and M.R.; data curation, N.K.; writing – original draft, C.C. and N.K.; writing – review & editing, B.P., C.C., and N.K.; supervision, B.P.; funding acquisition, B.P. and M.R.

## DECLARATION OF INTERESTS

The authors declare no competing interests.

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

## STAR★METHODS

### KEY RESOURCES TABLE

| REAGENT or RESOURCE | SOURCE | IDENTIFIER |
|---|---|---|
| **Deposited data** | | |
| Example Inputs for PIPEFISH Spatial Transcriptomics Pipeline Tool | This paper | https://doi.org/10.5281/zenodo.7909295 |
| *in situ* sequencing (ISS) of a whole coronal slice of a mouse brain | Garatic, M. et al. | https://doi.org/10.1101/2021.10.12.464086 |
| MERFISH of human U2-OS cell cultures | Moffitt, J.R. et al. | https://doi.org/10.1073/pnas.1612826113 |
| seqFISH of a developing mouse embryo | Lohoff, T. et al. | https://doi.org/10.1038/s41587-021-01006-2 |
| **Software and algorithms** | | |
| PIPEFISH, the HuBMAP Spatial Transcriptomics Pipeline | This Paper | https://doi.org/10.5281/zenodo.8170106 https://github.com/hubmapconsortium/spatial-transcriptomics-pipeline |

### RESOURCE AVAILABILITY

#### Lead contact
Further information and requests for data should be directed to and will be fulfilled by the lead contact, Cecilia Cisar (ccisar@ucsc.edu).

#### Materials availability
No materials were used in this study.

#### Data and code availability
- PIPEFISH, the original code used to generate these results, is publicly available on Github as of the date of publication. URL provided in the key resources table.
- Example data from each of the three datasets in this paper (seqFISH, ISS, and MERFISH) plus other necessary pipeline input files have been deposited at Zenodo. DOI is listed in the key resources table.
- Any additional information required to reanalyze the data reported in this paper is available from the lead contact upon request.

### METHOD DETAILS

#### Conversion to SpaceTx format
While there are a variety of published smFISH-based spatial transcriptomics assays, few have seen use outside their institution of origin.[29] This has resulted in the creation of many in-house codebases, oftentimes poorly documented and not understood by those outside that institution. To address this need, starfish was created as a universal FISH-analysis tool by offering a universal set of tools to format and extract information from all FISH experiment types. We have taken this one step further by simplifying the process of converting any lab's data into the format natively used by starfish, SpaceTx, by defining a small list of intuitive experiment-specific pipeline parameters that are used to parse the input data files. Image files with any number of dimensions and sorted in any way across folders can be parsed and formatted, so long as parameters are consistent between all files. Auxiliary views of images with non-FISH information, such as a DAPI view, can also be included in the conversion for downstream analysis, such as image registration or segmentation.

In some experiment types, it is common to use a 'pseudocolor' and 'pseudoround' indexing scheme to more efficiently pack barcodes into the experimentally available dye colors. This can offer computational advantages when computing the hamming distance between barcodes, as it can be assumed that there is only one valid pseudocolor channel among the imaging rounds and channels that comprise it.[30] By providing input parameters that describe how the experimental imaging rounds and channels correspond to the pseudocolors and pseudorounds used to define barcodes, the pipeline can automatically sort image channels into the desired pseudocolor scheme.

## CheckAll decoder

Starting from a set of spots of shape (*round*, *channel, z, y*, *x*), a codebook of barcodes of shape (*round*, *channel)* with no off rounds in any barcode, and a search radius value, the spots are connected into a non-overlapping set of barcodes that each match a barcode in the codebook. Two slightly different algorithms are used to balance the precision and recall. They share the same steps except the order of two of the steps are switched between the different versions. The following is for the "filter-first" version.

1. For each spot in each round, find all spot neighbors in other rounds that are within the search radius.
2. For each spot in each round, build all possible full length barcodes based on the channel labels of each spot's neighbors and itself.
3. Choose the "best" barcode of each spot's possible barcodes by calculating a score that is based on minimizing the spatial variance and maximizing the intensities of the spots in the barcode (shown below). Each spot is assigned a "best" barcode in this way.

$$Score \ = \ Q + Sv \ * \ C$$

$$Q \ = \ - \ log(1 \ / \ (1 + (RoundNum \ - \ QualSum))$$

$$RoundNum \ = \ Number \ of \ rounds \ in \ experiment$$

$$QualSum \ = \ Sum \ of \ normalized \ intensity \ values \ of \ all \ spots \ in \ barcode$$

$$Sv \ = \ -log(1 \ / \ 1 + (\Sigma_{i \ = \ x,y,z} \ var(coords_i)))$$

$$coords_i \ = \ Vector \ of \ spot \ coordinate \ values \ for \ all \ spots \ in \ the \ i^{th} \ dimension$$

$$C \ = \ constant \ (2 \ here)$$

4. Drop "best" barcodes that don't have a matching target in the codebook.
5. Only keep barcodes/targets that were found as "best" using at least *n* of the spots that make each up (*n* is determined by the mode parameter).
6. Find an approximate maximum independent set of the spot combinations so no two barcodes use the same spot.
   a. A graph is created where each node is a combination of spots that make up a decoded barcode and edges connect nodes that share at least one spot.
   b. Nodes are eliminated from the graph in order of highest number of edges to lowest, with ties being broken by choosing the barcode with the higher score (described in step three), until there are no longer any edges in the graph.

The other method, called "decode-first", is the same except steps 3 and 4 are switched so that the minimum scoring barcode is chosen from the set of possible codes that have a match to the codebook. The filter-first method will have lower recall but higher precision while the other method will have higher recall but at the cost of lower precision.

Decoding is run in multiple stages and the parameters change each stage to become less strict as it progresses. The high accuracy algorithm (filter-first) is always run first followed by the lower accuracy method (decode-first), each with slightly different parameters based on the choice of "mode" parameter. After each decoding, the spots found to be in decoded barcodes are removed from the original set of spots before they are decoded again with a new set of parameters. In order to simplify the number of parameters to choose from, we have sorted them into three sets of presets ("high", "medium", or "low" accuracy) determined by the "mode" decoding parameter.

Decoding is also done multiple times at different search radius values that start at 0 and increase incrementally until they reach the specified search radius. This allows high confidence barcodes to be called first which makes things simpler in subsequent stages as there are less spots to choose from.

## Article

If the error_rounds parameter is set to 1, after running all decodings for barcodes that exactly match the codebook, another set of decodings will be run to find barcodes that are missing a spot in exactly one round. If the codes in the codebook all have a hamming distance of at least 2 from all other codes, each can still be uniquely identified using a partial code with a single round dropped. Barcodes decoded with a error-corrected code like this are inherently less accurate and so an extra column was added to the final output table that labels each decoded target with the number of rounds that was used to decode it, allowing you to easily separate these less accurate codes from your high accuracy set if you wish.

## Quality control metrics

QC metrics are automatically computed in the pipeline given the output from prior steps. The values are both calculated on a per-FOV basis and for all combined FOVs, as described below.

Blank-based metrics (Figure 2) are key to understanding the accuracy of a decoded dataset. The values are calculated as follows.

1. Relative abundance of on-targets vs. off-targets: The total number of transcripts that match a given target are tallied. The sample average and standard deviation are calculated from the counts of off-target barcodes, and a two-tailed 95% confidence interval for the normal distribution with the same average and standard deviation are used to calculate the proposed cutoff for on-target barcodes. This CI is performed for both the uncorrected barcodes, and again for the combined corrected and uncorrected barcodes.
2. False positive rate: Optionally during the conversion process, off-target barcodes are inserted into the codebook that do not overlap within a specified hamming distance of existing codes. On a cell by cell basis, the count of transcripts and off-targets are tallied, then the FPR is calculated as $\frac{off-targets}{off-targets+transcripts}$.

Additional spot and transcript metrics can provide insight into the reason for low precision or recall (Figure S3). Spot-based metrics are not computed when the primary spot decoder is the PixelSpotDetector. Spots with no cell_id assigned from the segmentation mask will be filtered before these QC values are calculated.

Most transcript-based metrics are run on all datasets, regardless of which decoding method was used earlier in the pipeline. These statistics can offer more specific insight into how well a given dataset has been processed.

3. Spots per round: The number of spots in each round of the data is tallied. The standard deviation and skew of these counts are taken. If this is found to be increasing or decreasing across rounds, there is likely experimental error in the time between the collected images.
4. Spots per channel: The number of spots in each channel of the data is tallied. The standard deviation and skew of these counts are taken. If this is found to be significantly low or high in a particular channel, there may be issues with that fluorophore or with the channel normalization during image processing.
5. Transcript source spot counts across rounds: The rounds that were used to decode each transcript are inferred and then tallied. The standard deviation and skew of these counts are taken. If the codebook is redundant and spots are absent specific rounds, this could point to that particular round consistently being omitted due to experimental or processing issues.
6. Transcript source spot counts across channels: The channels that were used to decode each transcript are inferred and then tallied. The standard deviation and skew of these counts are taken. If the spots across channels are evenly distributed and the spots used for decoding are not, then this points to a problem in the decoding method. Alternatively, if the codebook is redundant, this could point to a particular channel consistently being omitted due to experimental or processing issues.

The above four metrics are also directly compared to each other in Figure S3A, where the normalized counts for both transcripts sources and spots are displayed on the same plot together.

7. Transcript count per cell: The number of transcripts assigned to each cell id are tallied. The cells with a count lower than $median - 1.5 \times (third\ quartile - first\ quartile)$ are suggested to be removed from analysis.
8. Spatial homogeneity of spots across each FOV:
   a. The AstroPy 'RipleyKEstimator' is used, configured to match image size.
   b. A total of 10 radii, $r$,i are evaluated on a range of 0 to $\sqrt{\frac{image\ area}{2}}$.
   c. The estimator is applied to both the spot data and a poisson process for each $r$.
   d. A 95% confidence interval for the null hypothesis (random poisson process) is generated by the Monte-Carlo method, sampling from a uniform random distribution over the same FOV size with the same number of spots.

If the spots are indeed non-random, the data should have a higher score (and thus be more clustered) than the 95% confidence interval for most calculated radii.

The following metrics are calculated automatically, but do not have a corresponding graph in the output pdf; they are exclusively in the human-readable.yml output.

9. Spots per barcode: Divide the total number of spots by the length of the codebook.

10. Fraction of spots used for transcripts: If a spot-based method was used, the number of spots that were used to decode the transcripts is taken. This is then divided by the total tally of spots found. Only applicable when an experiment has a redundant codebook. If barcodes are consistently using error correction, this suggests that either experimental protocol or processing/decoding parameters could be improved.

### Addition of noise for QC comparison

Multiplicative Gaussian noise was added to images to simulate images acquired with a shorter exposure time in Figure S2. Noise was added as $output = input + n * input$, where $n$ is a Gaussian random variable with mean=1.5 and variance=3.0.

### Pipeline parameters used for results

The individual parameter values used for each of the three example datasets to obtain the results shown here are specified in the metadata.json files found with the example data (Code and Data Availability).

### smFISH processing

smFISH images were corrected by subtracting dark-field images from each, applying a Gaussian high pass filter with a sigma value of 3, followed by deconvolution by the Richardson-Lucy method using a Gaussian point-spread-function with a sigma value of 3 for 15 iterations. Transcript locations were then identified using the *detect_spots* function from the FISH-quant Python package[31] using a voxel size of (4000, 110, 110) and a radius of 500. To calculate an optimal threshold for each transcript, spot finding was done in all FOVs at a range of threshold values and the total number of spots was plotted at each threshold. The elbow point of this curve was used as the final threshold for each transcript. The results were combined into a single expression table for all cells across all tiles for the 36 smFISH genes. The E8.5 mouse embryo 10x Genomics scRNA-seq data was downloaded using the MouseGastrulationData Bioconductor package. Cell type assignments for somitic and paraxial mesoderm types were amended with predictions from Guibentif et al.[32] (personal communication) and blood subtypes (erythroid 1, 2, and 3 and blood progenitors 1 and 2) were merged into two major groups. Cell types with less than 25 cells were then dropped. We then joined the seqFISH data expression matrices for all forty tiles, dropping any cells with fewer than 10 total transcripts or 5 unique transcripts.

The Tangram functions *map_cells_to_space* and *project_genes* were used in "clusters" mode to predict full genome expression in the seqFISH cells using the scRNA-seq counts. The predicted expression for the 36 genes measured by smFISH in each cell was compared to measured expression and the mean Pearson correlation coefficient calculated across all cells.

### Spot count curve as a noise metric

To show that the smFISH images used for the seqFISH external QC had varying levels of mixing between signal and noise intensities, we calculated the area under the curve of the normalized spot count vs. threshold curve. For each gene, the threshold and total count values were normalized by subtracting the respective minimum value from each and dividing by the maximum value (after subtracting the minimum). The area under the curve was then calculated using Simpson's method (scipy.integrate).

### Comparison of MERFISH results to bulk RNA-seq

The average count of each gene measured by MERFISH per cell was calculated and compared against the FPKM of the same gene measured by bulk RNA-seq in a sample of the same cell type. Bulk RNA-seq FPKMs were obtained from the original MERFISH authors (Jeffrey Moffit, personal communication). The Pearson correlation coefficient of the MERFISH and bulk RNA-seq FPKMs was then calculated and reported.

### Comparison of ISS results to reference atlas

Pipeline ISS images were created by setting transcript pixel locations to maximum brightness, dilating for 3 iterations (for visibility) and then adding the result to the DAPI image for each transcript. Atlas images were obtained from https://mouse.brain-map.org/ for each transcript that had a coronal section present in the database. The slice used was chosen by identifying the slice in the atlas images whose Nissl stain was most similar in morphology to the ISS dataset's DAPI stain (both nucleus stains). The "Expression" and "ISH" image for that atlas slice were downloaded and merged and then cropped and rotated to match the pipeline ISS image.

