## [Document S2. Transparent peer review record for Cecilia Cisar et al · Cell Genomics]

A Unified Pipeline for FISH Spatial Transcriptomics

Cecilia Cisar^{1,2,3,4}, Nicholas Keener^{2,3}, Mathew Ruffalo⁵, Benedict Paten³

Summary

Initial submission: Received : 2/21/2023

Scientific editor: Laura Zahn

First round of review: Number of reviewers: 2
Revision invited : 4/3/2023
Revision received : 6/3/2023

Second round of review: Number of reviewers: 2
Accepted : 7/24/2023

Data freely available: Yes

Code freely available: Yes

This transparent peer review record is not systematically proofread, type-set, or edited. Special characters, formatting, and equations may fail to render properly. Standard procedural text within the editor's letters has been deleted for the sake of brevity, but all official correspondence specific to the manuscript has been preserved.

Referees' reports, first round of review

Reviewer #1:

Cisar et al. present in their manuscript titled "A Unified Pipeline for FISH Spatial Transcriptomics" an image analysis pipeline for analyzing ⁶FISH-based spatial transcriptomics experiments. The pipeline extends the starfish library with a new data conversion script and decoding algorithm and adds a quality assessment

step to the workflow. I agree with the authors that a unified pipeline for the FISH-based spatial transcriptomic community is missing, and I was excited to try out their solution. Unfortunately, while the authors imply that their pipeline is also directed at labs without a dedicated computational scientist, I encountered several errors while trying to install and run the pipeline due to the very few installation instructions. While using a docker image to circumvent version conflicts of different libraries is potentially a great solution, more detailed install instructions are needed. Equally, I was missing a detailed tutorial of all steps and a more detailed explanation of each step's parameters and expected outcomes. The integration of quality control metrics is an excellent addition to the pipeline! The false-positive barcodes are a simple but great tool. The CheckAll Decoder is an exciting method to maximize the detection of barcodes in the samples. The method gives promising results while, unfortunately, increasing run time and memory use quite extensively.

Major Comments:

- Add clear installation instructions (including for M1 Mac silicon).
- While the GitHub repo explains each parameter, a more extensive tutorial and walkthrough, possibly using a test dataset, are missing. The list of parameters that must be set is quite long, and it is not directly clear how each parameter affects the pipeline performance. Additionally, adding expected outcomes for each step would increase the usability of the pipeline.
- I was surprised that the authors did not integrate state-of-the-art cell segmentation algorithms like cellpose or stardist. While the authors state that cell masks can be created externally and uploaded to the pipeline, the point of a pipeline is to have fewer external steps that need to be performed outside the pipeline. While there are undoubtedly tricky samples where even cellpose or stardist won't give the best result, integrating those models into the pipeline would be an excellent advantage for the workflow.
- I'm also unsure how representative the simulated nuclear images used for benchmarking are. Looking at Figure S4, the simulated images would only represent S4c. I would encourage the authors to use a more complex dataset (simulated or real with GT annotations) to benchmark their segmentation approach.

Minor Comments:

- In the introduction, the authors discuss the development of spatial transcriptomic methods but fail to cite some of the key papers in the field like:
 - o Lubeck, E., Coskun, A., Zhiyentayev, T. et al. Single-cell in situ RNA profiling by sequential hybridization. *Nat Methods* 11, 360-361 (2014)
 - o Eng, CH.L., Lawson, M., Zhu, Q. et al. Transcriptome-scale super-resolved imaging in tissues by RNA seqFISH+. *Nature* 568, 235-239 (2019).
 - o Femino, A. M., Fay, F. S., Fogarty, K. & Singer, R. H. Visualization of single RNA transcripts in situ. *Science* 280, 585-590 (1998).
 - o Raj, A., van den Bogaard, P., Rifkin, S. A., van Oudenaarden, A. & Tyagi, S. Imaging individual mRNA molecules using multiple singly labeled probes. *Nature Methods* 5, 877-879 (2008).
 - o Lee, J. H. et al. Highly multiplexed subcellular RNA sequencing in situ. *Science* 343, 1360-1363 (2014).
- Figure S3 needs to be better described in the Quality control metrics section and is hard to understand and a bit hard to read with the size difference of the labels.
- The order of the sections "Quality control metrics", "Pipeline performance.." and "Internal quality check" is a bit confusing. Restructuring this part would help the flow of the paper and help the reader understand the QC metrics.
- I would repeat a short definition of the term "error-corrected" in the QC section to increase understanding in this section.
- In some parts, the false-positive barcodes are referred to as "blanks" and in other parts as "off-target". Sticking to one label here can increase the understanding of this section.
- Figure S5 needs to be better explained in the text or methods.
- Something probably went wrong during the conversion of the files, but several supplemental figures are cropped at the bottom. Also, Figure S5 needs to be easier to read with overlapping figure elements.
- Page 12: Incomplete sentence: "We obtained FPKM measurements calculated from bulk RNA-seq experiments of the same cell type as used in the MERFISH images²⁴ (personal communication) and compared these values to the average expression of the cells in the MERFISH images and found them to be highly correlated with a Pearson correlation coefficient of 0.85 (Methods) (Figure 3b), indicating that the counts the pipeline produced are similar to those obtained by."

Reviewer #2: Cisar and colleagues have developed a new tool called PIPEFISH for smFISH analysis of many targets simultaneously and transcript annotation. Current methods for processing and analyzing this type of data are challenging without computational capabilities in the laboratory because much of the software available for analysis is in-house code with little documentation. Publicly available, open-source tools for analysis will make this data more accessible to other labs.

The authors benchmark their method by showing that "on-target" barcodes are found at a higher frequency than blank "off-target" codes in seqFISH, MERFISH and ISS. They go on to validate the results of their analysis pipeline by comparing the predicted gene expression with the measured expression of the same gene by smFISH, bulk RNA seq and reference expression of the same genes in similar brain slices from the Allen Brain Atlas.

I have a few suggestions for improvement of the manuscript:

For figure 3a, the authors report an average Pearson correlation of 0.47 and the image is showing the top 6 genes but even the top 6 genes are showing a Pearson correlation ranging from 0.754 to 0.859. It makes sense that this data is noisy due to the single-cell differences but it would be helpful if the authors could expand on what information could be extracted from this comparison.

The sentence describing figure 3b seems to be missing a word at the end:

"We obtained FPKM measurements calculated from bulk RNA-seq experiments of the same cell type as used in the MERFISH images²⁴ (personal communication) and compared these values to the average expression of the cells in the MERFISH images and found them to be highly correlated with a Pearson correlation coefficient of 0.85 (Methods) (Figure 3b), indicating that the counts the pipeline produced are similar to those obtained by ."

For Figure 3a and 3c, it would be helpful to include a legend or a description of the colors.

I appreciate the goal of this manuscript, which is to enable non-computational researchers to perform this type of analysis. The authors have done a very nice job with a well-annotated pipeline for running the analysis. However, if the goal of the authors is to enable labs without computational members, then some additional information would be helpful for users, which I have listed here (to be clear, I am not requiring these, just writing something that I think would help a non-computational lab):

1. Since the input must be accepted as a tiff file--it would be helpful to include options for converting all of the common file formats to tiff format (for example, nd2 to tiff)
2. Although the method is well-annotated, if the goal is for non-computational labs to run this, then a section with step-by-step instructions for how to run the commands (even a screenshot showing a simple command from a mac or a PC would be extremely beneficial)

Authors' response to the first round of review

Reviewer #1:

Cisar et al. present in their manuscript titled "A Unified Pipeline for FISH Spatial Transcriptomics" an image analysis pipeline for analyzing FISH-based spatial transcriptomics experiments. The pipeline extends the starfish library with a new data conversion script and decoding algorithm and adds a quality assessment step to the workflow. I agree with the authors that a unified pipeline for the FISH-based spatial transcriptomic community is missing, and I was excited to try out their solution. Unfortunately, while the authors imply that their pipeline is also directed at labs without a dedicated computational scientist, I encountered several errors while trying to install and run the pipeline due to the very few installation instructions. While using a docker image to circumvent version conflicts of different libraries is potentially a great solution, more detailed install instructions are needed. Equally, I was missing a detailed tutorial of all steps and a more detailed explanation of each step's parameters and expected outcomes. The integration of quality control metrics is an excellent addition to the pipeline! The false-positive barcodes are a simple but great tool.

The CheckAll Decoder is an exciting method to maximize the detection of barcodes in the samples. The method gives promising results while, unfortunately, increasing run time and memory use quite extensively.

Major Comments:

- Add clear installation instructions (including for M1 Mac silicon).

Installation instructions on the github have been elaborated on significantly. Usage instructions should be the same for a M1/2-based Mac, however we did not realize that our docker image was incompatible with ARM-based systems. We have built a multi-platform build (despite a related bug in Starfish) and verified that PIPEFISH runs to completion on an ARM-based Mac. This is integrated to our repository for the primary image of the pipeline (starfish-custom), and we will work on developing this for optional features (cellpose, baysor) as well.

- While the GitHub repo explains each parameter, a more extensive tutorial and walkthrough, possibly using a test dataset, are missing. The list of parameters that must be set is quite long, and it is not directly clear how each parameter affects the pipeline performance. Additionally, adding expected outcomes for each step would increase the usability of the pipeline.

We have added directions on how to run the example datasets that we have documented in this paper, and starting from these examples gives the user input yml and json files that they can build off of. We have also added a flowchart that illustrates the various pipeline control variables, as well as brief descriptions of the expected output from each step.

- I was surprised that the authors did not integrate state-of-the-art cell segmentation algorithms like cellpose or stardist. While the authors state that cell masks can be created externally and uploaded to the pipeline, the point of a pipeline is to have fewer external steps that need to be performed outside the pipeline. While there are undoubtedly tricky samples where even cellpose or stardist won't give the best result, integrating those models into the pipeline would be an excellent advantage for the workflow.

Thanks, this was a good suggestion. We've integrated the CellPose segmentation tool into PIPEFISH as a segmentation option and now users don't need to use outside tools to obtain high quality cell segmentations. Users can choose any of CellPose's pretrained segmentation models or can supply their own custom model file in order to get the best possible results. We have updated all the result figures in the paper using segmentations from CellPose which slightly improved the performance according to our quality control metrics.

- I'm also unsure how representative the simulated nuclear images used for benchmarking are. Looking at Figure S4, the simulated images would only represent S4c. I would encourage the authors to use a more complex dataset (simulated or real with GT annotations) to benchmark their segmentation approach.

As we are now using CellPose as our primary segmentation tool, we are dropping the benchmarking analysis from the paper since CellPose is an already well-tested and proven tool.

Minor Comments:

- In the introduction, the authors discuss the development of spatial transcriptomic methods but fail to cite some of the key papers in the field like:

o Lubeck, E., Coskun, A., Zhiyentayev, T. et al. Single-cell in situ RNA profiling by sequential hybridization. Nat Methods 11, 360-361 (2014)

o Eng, C.H.L., Lawson, M., Zhu, Q. et al. Transcriptome-scale super-resolved imaging in tissues by RNA seqFISH+. Nature 568, 235-239 (2019).

o Femino, A. M., Fay, F. S., Fogarty, K. & Singer, R. H. Visualization of single RNA transcripts in situ. Science 280, 585-590 (1998).

o Raj, A., van den Bogaard, P., Rifkin, S. A., van Oudenaarden, A. & Tyagi, S. Imaging individual mRNA molecules using multiple singly labeled probes. Nature Methods 5, 877-879 (2008).

o Lee, J. H. et al. Highly multiplexed subcellular RNA sequencing in situ. Science 343, 1360-1363 (2014).

Thanks, we have added the suggested citations to the paper and revised the introduction accordingly.

- Figure S3 needs to be better described in the Quality control metrics section and is hard to understand and a bit hard to read with the size difference of the labels.

The following has been added in the inline reference to the figure (now S2): "...as the false positive rates and the threshold of true decoded barcodes that can be delineated from false-positive barcodes give a quantifiable point of comparison between datasets."

- The order of the sections "Quality control metrics", "Pipeline performance.." and "Internal quality check" is a bit confusing. Restructuring this part would help the flow of the paper and help the reader understand the QC metrics.

The intent of these sections was to transition from describing PIPEFISH as a whole and all the steps therein (the last of which was QC) to describing the results of running PIPEFISH on experimental data, which we evaluate with QC metrics. We have changed some of the language around these sections to make this transition more apparent.

- I would repeat a short definition of the term "error-corrected" in the QC section to increase understanding in this section.

We've added a short recap of what we mean by error-corrected in the "Quality Control Metrics" section.

- In some parts, the false-positive barcodes are referred to as "blanks" and in other parts as "off-target". Sticking to one label here can increase the understanding of this section.

We replaced all instances of referring to false-positive barcodes as "blanks" with "off-target" to be more consistent.

- Figure S5 needs to be better explained in the text or methods.

Explanations for possible uses of the metrics represented in this figure (now S4) have been added to the methods section.

- Something probably went wrong during the conversion of the files, but several supplemental figures are cropped at the bottom. Also, Figure S5 needs to be easier to read with overlapping figure elements.

Fixed the overlapping elements in S5 (now S4) and ensured nothing got cut-off in conversion. Thank you.

- Page 12: Incomplete sentence: "We obtained FPKM measurements calculated from bulk RNA-seq experiments of the same cell type as used in the MERFISH images²⁴ (personal communication) and compared these values to the average expression of the cells in the MERFISH images and found them to be highly correlated with a Pearson correlation coefficient of 0.85 (Methods) (Figure 3b), indicating that the counts the pipeline produced are similar to those obtained by ."

Fixed this. Thank you.

Reviewer #2: Cisar and colleagues have developed a new tool called PIPEFISH for smFISH analysis of many targets simultaneously and transcript annotation. Current methods for processing and analyzing this type of data are challenging without computational capabilities in the laboratory because much of the software available for analysis is in-house code with little documentation. Publicly available, open-source tools for analysis will make this data more accessible to other labs.

The authors benchmark their method by showing that "on-target" barcodes are found at a higher frequency than blank "off-target" codes in seqFISH, MERFISH and ISS. They go on to validate the results of their analysis pipeline by comparing the predicted gene expression with the measured expression of the same gene by smFISH, bulk RNA seq and reference expression of the same genes in similar brain slices from the Allen Brain Atlas.

I have a few suggestions for improvement of the manuscript:

For figure 3a, the authors report an average Pearson correlation of 0.47 and the image is showing the top 6 genes but even the top 6 genes are showing a Pearson correlation ranging from 0.754 to 0.859. It makes sense that this data is noisy due to the single-cell differences but it would be helpful if the authors could expand on what information could be extracted from this comparison.

We have done some digging into the reason for the large range of correlations between the predicted counts and the actually measured smFISH counts and believe this to be a data quality issue with the smFISH images. We have added additional text and a new supplementary figure (S8) to show this:

"The large range of correlation coefficients (0.12 - 0.85) is likely a result of differing image qualities between the smFISH genes. As the smFISH images are not multiplexed, they are very sensitive to the choice of threshold when identifying spots in the images. Thresholds for each gene were determined by plotting the number of spots found at a large range of thresholds and calculating the elbow point of the curve. The area under the curve (AUC) can also be used as a rough estimate of how much the intensities of the true fluorescent signal and the noise in an image overlap and large overlaps would make separating signal from noise impossible. We found that the correlation between predicted and smFISH counts to be fairly strongly correlated with the AUC of the spot count vs threshold curve with a Pearson correlation of -0.67 (Figure S8 (included below for convenience)), indicating that the genes where performance is low are likely a result of poor separation between signal and noise intensities in that image and not poor quality seqFISH counts."

Figure S8: Explanation of poorly performing genes for seqFISH external QC

Normalized total spot vs threshold curves for smFISH images for a) Pcgf1 and b) Basp1. R^2 printed is the Pearson correlation between predicted and smFISH counts for that gene while the area is the integral of the normalized total spot vs threshold curve. The red highlighted point indicates the elbow point of the curve that was used as the threshold value to obtain the printed Pearson correlation. c) Pearson correlation of predicted and smFISH counts vs the area under the normalized spot count curve for all genes. R^2 printed is the Pearson correlation between the Pearson correlation of predicted and smFISH counts and the area under the normalized spot count curve.

The sentence describing figure 3b seems to be missing a word at the end:

"We obtained FPKM measurements calculated from bulk RNA-seq experiments of the same cell type as used in the MERFISH images²⁴ (personal communication) and compared these values to the average expression of the cells in the MERFISH images and found them to be highly correlated with a Pearson correlation coefficient of 0.85 (Methods) (Figure 3b), indicating that the counts the pipeline produced are similar to those obtained by ."

Fixed this. Thank you.

For Figure 3a and 3c, it would be helpful to include a legend or a description of the colors.

We have added a legend for 3a and added additional description for 3c.

I appreciate the goal of this manuscript, which is to enable non-computational researchers to perform this type of analysis. The authors have done a very nice job with a well-annotated pipeline for running the analysis. However, if the goal of the authors is to enable labs without computational members, then some additional information would be helpful for users, which I have listed here (to be clear, I am not requiring these, just writing something that I think would help a non-computational lab):

1. Since the input must be accepted as a tiff file--it would be helpful to include options for converting all of the common file formats to tiff format (for example, nd2 to tiff)

While PIPEFISH is intended to be a universal tool for FISH-based ST, we cannot necessarily compensate for the variety of proprietary formats available for imaging tools. Additionally, in many cases we cannot easily integrate it into the pipeline due to a lack of command-line based conversion tools. To address this, we have added a paragraph to the README introduction that points users to the openly available Bioformats to convert input images if need be.

2. Although the method is well-annotated, if the goal is for non-computational labs to run this, then a section with step-by-step instructions for how to run the commands (even a screenshot showing a simple command from a mac or a PC would be extremely beneficial)

We have added more thorough directions on how to run the example datasets that we have documented in this paper, and had them tested by someone less experienced with computational tools.

Referees' reports, second round of review

Reviewer #1: The authors addressed my comments in their revised manuscript/online resources. However, a few points remain to be addressed/clarified:

The installation instructions and example datasets provide an easier start for running the pipeline. However, I'm still not convinced that "labs without computational members" will find it easy to run the pipeline with the current description. I would either adjust the language in the introduction or include even more instructions on installing and running the pipeline.

I was able to run the first two example datasets but failed to run the third example. I suspect that I ran out of memory for this dataset. Could the authors add the memory requirements to running these datasets so users can estimate if a workstation or server is needed for their analysis?

I found the pdf with the quality controls helpful, but it was a bit hard to find the right description in the method section for each of the plots. I think it would be helpful to keep the title of the plots consistent with the description and the order in the method section.

Reviewer #2: Comments enter in this field will be shared with the author; your identity will remain anonymous.

Authors' response to the second round of review

Reviewers' Comments:

Reviewer #1: The authors addressed my comments in their revised manuscript/online resources. However, a few points remain to be addressed/clarified:

The installation instructions and example datasets provide an easier start for running the pipeline. However, I'm still not convinced that "labs without computational members" will find it easy to run the pipeline with the current description. I would either adjust the language in the introduction or include even more instructions on installing and running the pipeline.

We have adjusted the language in the introduction and removed the phrase "labs without computational members".

I was able to run the first two example datasets but failed to run the third example. I suspect that I ran out of memory for this dataset. Could the authors add the memory requirements to running these datasets so users can estimate if a workstation or server is needed for their analysis?

We have added memory requirements for all datasets to both the github repo under the "Example PIPEFISH Run" heading and on the zenodo page where the datasets are downloaded from.

I found the pdf with the quality controls helpful, but it was a bit hard to find the right description in the method section for each of the plots. I think it would be helpful to keep the title of the plots

consistent with the description and the order in the method section.

We have rearranged the order of the metrics in the methods section to match the order they are displayed in the Figure S3 plots. Additionally, we have changed the names of each numbered list item in QC methods to match the titles in the Figure S3 subplots.

Reviewer #2: Comments enter in this field will be shared with the author; your identity will remain anonymous.